# Momentum Capsule Networks

**Josef Gugglberger**                                                    *j.gugglberger@gmail.com*
*Department of Computer Science*
*University of Innsbruck, Austria*

**David Peer**                                                           *david.peer@deepopinion.ai*
*DeepOpinion*

**Antonio Rodríguez-Sánchez**                         *antonio.rodriguez-sanchez@uibk.ac.at*
*Department of Computer Science*
*University of Innsbruck, Austria*

**Reviewed on OpenReview:** *https://openreview.net/forum?id=Su29OsknyQ*

## Abstract

Capsule networks are a class of neural networks that aim at solving some limiting factors of Convolutional Neural Networks. However, baseline capsule networks have failed to reach state-of-the-art results on more complex datasets due to the high computation and memory requirements. We tackle this problem by proposing a new network architecture, called Momentum Capsule Network (MoCapsNet). MoCapsNets are inspired by Momentum ResNets, a type of network that applies reversible residual building blocks. Reversible networks allow for recalculating activations of the forward pass in the backpropagation algorithm, so those memory requirements can be drastically reduced. In this paper, we provide a framework on how invertible residual building blocks can be applied to capsule networks. We will show that MoCapsNet beats the accuracy of baseline capsule networks on MNIST, SVHN, CIFAR-10 and CIFAR-100 while using considerably less memory. The source code is available on `https://github.com/moejoe95/MoCapsNet`.

## 1 Introduction

Deep neural networks have been able to achieve impressive results on many computer vision tasks. Neural networks, like AlexNet (Krizhevsky et al., 2012), VGG nets (Simonyan & Zisserman, 2014) or ResNets (He et al., 2016), consist mostly of a combination of many convolutional and pooling layers. One limiting factor of CNNs is their inability to learn viewpoint invariant features. A CNN aimed at image classification is likely to misclassify an image of an upside-down flipped object if the training data only contained images of the object from other orientations. Capsule networks were designed to overcome this drawback (Sabour et al., 2017). Neurons in a capsule define the properties of an object, while its length relates to the presence of such object. Instead of the pooling operations used in most well-known CNNs, capsule networks make use of dynamic routing algorithms. This routing procedure is superior to (max-) pooling because pooling throws away valuable information by only taking the most active feature detector into account (Sabour et al., 2017).

Unfortunately, capsule networks suffer from their own limitations. First, capsule networks contain a large number of trainable parameters, because it is necessary for each capsule in layer $l$ to compute a vote for each capsule in layer $l + 1$, which leads to a memory consumption that grows quadratically with the number of capsules per layer. Therefore, such capsule networks are very resource intensive, and their memory requirements are a bottleneck that prevents possibilities for training deeper architectures, and as such, limit their usability. Second, deep capsule networks suffer from instabilities, which can be overcome by training deep capsule networks using identity shortcut connections between capsule layers, allowing capsule networks with a depth of up to 16 capsule layers (Gugglberger et al., 2021).

For CNNs, the depth of the network is of great importance when maximizing the performance of such neural networks. The bottleneck for training deeper networks is memory, as memory requirements are proportional to the number of weights in a neural network. One possible way to overcome this bottleneck is to trade memory for computation by turning blocks of residual networks into reversible functions (Sander et al., 2021). We introduce in this work a way to include invertible residual blocks for the case of capsule networks. The proposed approach is also compatible with different routing algorithms, as we will demonstrate in the experimental section. Similar to Momentum ResNets, we use a momentum term that enables us to convert any existing residual capsule block into its reversible counterpart. Our architecture, called MoCapsNet, drastically reduces the memory consumption of deep capsule networks, such that we can train capsule networks at almost any arbitrarily deep configuration. A classical block of a residual capsule network increases the memory footprint of a model by around 185 MB, while our block adds just around 4 MB. We further show that MoCapsNet outperforms the performance of baseline capsule networks on MNIST, SVHN, CIFAR-10 and CIFAR-100 with accuracies of 99.54%, 93.00%, 72.18% and 43.48% compared to the baseline values of 99.36%, 92.13%, 71.49% and $\approx 1\%$. Our model is also able to overcome the instabilities deep capsule networks face during training to some degree. We show this by training a deep capsule network with 40 layers.

## 2 Related Work

### 2.1 Capsule Networks

The key idea of capsule networks is derived from computer graphics, where during the rendering process, images are generated from data structures. Capsule networks are constructed to invert rendering, meaning that they perform a conversion from an image to certain instantiation parameters. During inference, the parameters of a capsule represent an object or part of an object and therefore, much like an inverse computer graphics approach. These parameters are organized in vector form, called a *capsule*, where each element would correspond to a neuron in the capsule neural network. Capsules of a lower layer *vote* for the orientation of capsules in the upper layer by multiplying their representing vectors with a transformation matrix. Such transformation matrices are obtained through training, and they encode viewpoint-invariant part-whole relationships. Routing algorithms are aimed at computing the agreement between two capsules of subsequent layers, where a high value represents strongly agreeing capsules. The computed agreement is a scalar value that weights the votes from the lower level capsules and decides where to *route* the output of a capsule. Sabour et al. (2017) proposed the first capsule network with dynamic routing, reaching state-of-the-art results on the MNIST dataset. Later, Hinton et al. (2018) published *Matrix Capsules*, a capsule network with a more powerful routing algorithm based on expectation-maximization and reported a new state-of-the-art performance on the Small-NORB dataset. Many variants of capsule networks have been proposed in recent years (Ribeiro et al., 2020; Xiang et al., 2018; Yang et al., 2020; Chang & Liu, 2020; Ding et al., 2019; Sun et al., 2021; Peer et al., 2018), but for many tasks (i.e., CIFAR-10, CIFAR-100 or ImageNet) even the most recent approaches (Ribeiro et al., 2020) provide much higher error rates the than state-of-the-art CNN approaches (i.e., 9% on CIFAR-10, while CNNs provide errors of just 3% (Wistuba et al., 2019)).

### 2.2 Reversible Architectures

A network is called reversible if one can recalculate all its activations in the backward pass. One advantage of invertible networks is that they can perform backpropagation without saving the activations from the forward pass, which largely decreases the memory footprint of models where such a strategy is applied. Recently, many reversible or partly reversible architectures have been proposed, like RevNet (Gomez et al., 2017) or the three reversible alternatives to ResNets (Chang et al., 2018). Sander et al. (2021) introduced a way of turning any existing ResNet into a Momentum ResNet – without the need to change the network – by making the building blocks of residual neural networks reversible. Momentum ResNets change the forward rule of a classic residual network by inserting a momentum term ($\gamma$). Momentum ResNets would then be a generalization of classical ResNets (for $\gamma = 0$) and RevNets (where $\gamma = 1$). Additionally, ResNets using Momentum have a higher degree of expressivity than classical residual networks.

### 2.3 Residual Networks

Very deep convolutional neural networks have shown impressive results in many computer vision tasks. For example, VGG-net (Simonyan & Zisserman, 2014) provided an impressive error rate of just 23.7% on the ILSVRC-2014 competition, compare this result with the 38.1% error of the more shallow AlexNet (Krizhevsky et al., 2012), even though both networks consisted of the same building blocks. Unfortunately, as networks get deeper, the vanishing gradient problem (Hochreiter, 1991) becomes increasingly prevalent, which means that at some point stacking up more layers will lead to a drop in training accuracy. The vanishing gradient problem can be compensated to some degree by normalization techniques, such as batch normalization (Ioffe & Szegedy, 2015), or special weight initialization (Glorot & Bengio, 2010). Another way to improve gradient flow through the network is to add shortcut connections, connecting the output of a layer to the output of a layer that is deeper into the network (He et al., 2016), this approach is known as the *deep residual learning framework*. The possibility to skip layers stabilizes training and allows the training networks with more than 1000 layers (He et al., 2016). Residual learning can be applied to capsule networks as well (Gugglberger et al., 2021) such that the training of deep capsule networks can be stabilized by using identity shortcut connections between capsule layers, which allows training capsule networks with a depth of up to 11 layers when RBA (Sabour et al., 2017) is the routing algorithm and up 16 layers for SDA (Peer et al., 2018) and EM (Hinton et al., 2018) routing.

## 3  Momentum Capsule Networks

Expressivity grows exponentially with network depth in the case of CNNs, making it one of its most important hyperparameters (Raghu et al., 2017). One limiting factor that needs to be addressed is the memory bottleneck: common machine learning frameworks such as PyTorch (Paszke et al., 2019) implement automatic reverse mode differentiation algorithms [1] to compute the gradient. For non-invertible intermediate layers, it is necessary to cache all output values during forward propagation in order to compute the gradient correctly (Gomez et al., 2017). As capsules are multidimensional, they require significantly more memory during training when compared to classical neural networks or CNNs.

Reversible architectures reduce memory consumption by allowing to recalculate neuron activations in the backward pass instead of saving them in the forward pass (Gomez et al., 2017). In this work, we introduce the Momentum Capsule Network, or MoCapsNet for short. We will show how we can design capsule networks, such that a part of the network can be inverted. Following previous work on Momentum ResNets (Section 2.2), we will use a momentum term to make the blocks of a residual network invertible. In our MoCapsNet, a block will contain capsule layers instead of convolutional layers, which involves additional complexity when compared to Momentum ResNets.

An overview of the architecture is shown in Figure 1. MoCapsNet is implemented as follows: After the first convolutional layer, we perform another convolution and reshape it into capsules to form the *PrimaryCapsules* layer. Between *CapsLayer 1* and *CapsLayer 2*, the $n$ momentum residual capsule blocks (gray box in Figure 1) are inserted, each consisting of two capsule layers and shortcut connections, where the residual capsule blocks come with a modification of the forward rule. The smallest MoCapsNets consist of one residual block, as each block is composed of 2 hidden capsule layers (Figure 2) and these are added to the classical Capsule Networks. The smallest MoCapsNet will consist of four fully connected capsule layers: First capsule layer, 1 block (= 2 hidden capsule layers) and output capsule layer (as in Figure 1). We will explain the details on the momentum residual capsule blocks and the forward rule in Section 3.1. In order to implement the shortcut connections, an element-wise addition over the capsule votes after the squashing non-linearity is performed. As we add the capsule votes from a previous layer to the votes of the current layer, the addition of the shortcut connections happens after the routing process is already completed, and as such, it is totally independent of the number of routing iterations. The dimensions between the capsule layers do not change, the shortcut connection does not contain any learnable parameters. In the diagram of Figure 1 *CapsLayer 2*, has one capsule for each class. Between capsule layers, we perform dynamic routing for 3 routing iterations. The reconstruction network is made up of three dense layers, where the first two layers use *ReLu*, and

---

[1]Automatic Differentiation Package, `https://pytorch.org/docs/stable/autograd.html`, Accessed 01/2022

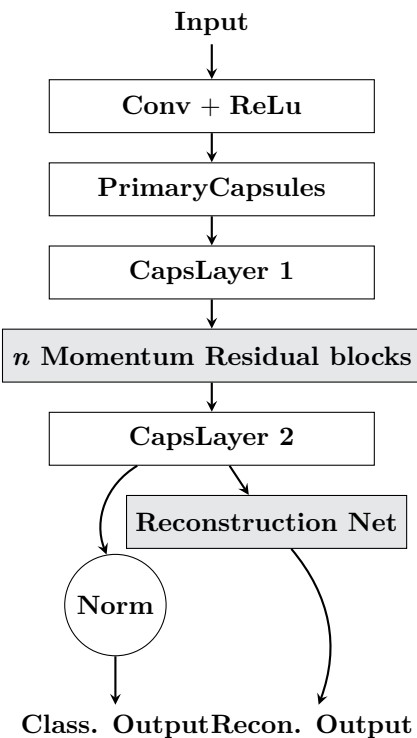

Figure 1: A high-level overview of our capsule network. Momentum residual blocks are shown in Figure 2b.

the last layer implements a *sigmoid* activation function. The reconstruction network is used to compute the *reconstruction loss* (Sabour et al., 2017) in the training procedure of the capsule network. The reconstruction loss corresponds to the sum of squared differences between the input pixels of the image and the output from the reconstruction network. The overall loss is the sum of the *margin* loss $L_{margin}$ (Sabour et al., 2017) and the reconstruction loss $L_{recon}$, which is weighted by a scalar factor $\lambda$:

$$L = \lambda L_{recon} + L_{margin} \tag{1}$$

The margin loss ensures that the vector representing a capsule is large, if and only if the object that the capsule should represent is present in the input image. In our experiments, we set $\lambda = 5 * 10^{-4}$.

### 3.1   Momentum Residual blocks

The residual building block for capsule networks (Gugglberger et al., 2021) can be seen in Figure 2a. The design of the block is very similar to a common residual building block used in ResNet architectures, but such blocks need to be adapted in order to handle capsule layers. This is because they represent capsules (not neurons), which perform dynamic routing with upper level capsules. The layers $f_{caps1}$ and $f_{caps2}$ shown in Figure 2a and Figure 2b are fully connected capsule layers with dynamic routing, as introduced by Sabour et al. (2017). To insert a shortcut connection, the input of the residual block is added to the output of the first and second capsule layers by an element-wise addition on the capsule votes. Different from the classical ResNet architecture, the addition is performed after the non-linearity (Figure 2a), which is a squashing function (Sabour et al., 2017).

Momentum residual blocks (Figure 2b) follow from Momentum ResNets (Sander et al., 2021), which are a modification of classical residual networks. To be able to use momentum, the forward rule of the network needs to be changed in order to make the residual blocks invertible. Thanks to this new formulation, we

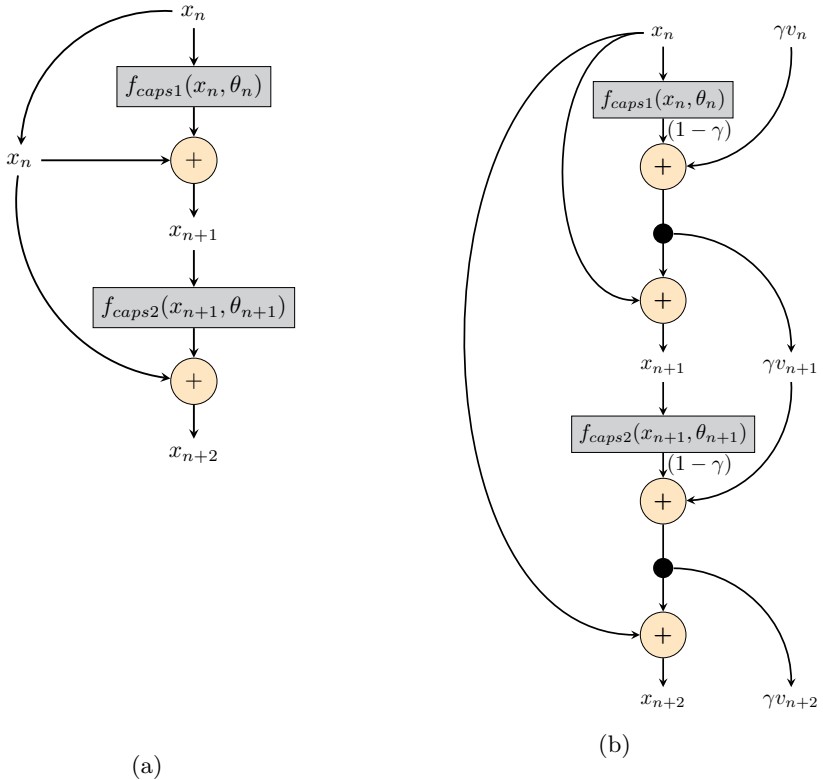

(a)

(b)

Figure 2: Residual building blocks for ResCapsNet (a) and MoCapsNet (b). The gray-colored elements correspond to capsule layers, and the orange-colored elements represent the addition of the shortcut connections.

are able to recalculate the activations of a block in the backward pass instead of having to save them into memory. The forward rule of a classical residual building block is:

$$x_{n+1} = x_n + f(x_n, \theta_n), \tag{2}$$

where $f$ is a function parameterized by $\theta_n$, and $x_n$ is the output of the previous layer. Invertibility is achieved by changing this forward rule into a velocity formulation ($v_n$), which introduces a momentum term $\gamma \in [0, 1]$, defined as:

$$v_{n+1} = \gamma v_n + (1 - \gamma) f(x_n, \theta_n), \tag{3}$$

Making use of this velocity, the new forward rule for Momentum capsule networks is defined as:

$$x_{n+1} = x_n + v_{n+1}. \tag{4}$$

Both forward rules can be seen in Figure 2, where the original forward rule used in residual capsule networks (Figure 2a) can be compared to the forward rule used in momentum capsule networks (Figure 2b).

Equations 3 and 4 can now be inverted, the latter is inverted as follows:

$$x_n = x_{n+1} - v_{n+1}, \tag{5}$$

while equation 3 is inverted as:

$$v_n = \frac{1}{\gamma}(v_{n+1} - (1 - \gamma)f(x_n, \theta_n)). \tag{6}$$

For the sake of clarity, we describe the general forms of the forward and backward steps in algorithms 1 and 2 respectively. In the forward pass, the first step is to initialize the velocity tensor $v$, for example by setting all values to zero (line 2 of Algorithm 1). Next, we iterate over the residual layers and compute the velocity and the output of each layer (Algorithm 1, lines 4-6). To recompute the activations in the backward pass, we only need the velocity and activations of the very last layer by saving their corresponding tensors (line 7 of Algorithm 1).

For the backward pass, we load the output and the velocity of the last layer (line 2 of Algorithm 2). In line 3 we initialize the variable for the gradient of the velocity. We then iterate over the residual layers to re-compute the activations and velocities backwards from the last to the first layer (Algorithm 2, lines 4-9). Finally, the gradients are computed as in line 10 of Algorithm 2. Because we can re-compute the neuron activations, we do not need to save them in the forward pass. This process makes the memory requirements for deep residual networks much reduced, which allows the training of very deep capsule networks. Our experimental evaluation also shows that the reduction of memory consumption shrinks at much larger rates than the increase in training time, which further motivates the advantage of the proposed approach.

Algorithm 1: High level Python code of the modified forward step of a momentum capsule network. The current inputs along with the residual layers are passed into the forward function below.

```
1  def forward(x, layers, gamma=0.9):
2      v = initialize()
3      for layer in layers:
4          v *= gamma
5          v += (1 - gamma) * layer(x)
6          x = x + v
7      save(x, v) # save for backward step
8      return x
```

Algorithm 2: High level Python code of the modified backward step of a momentum capsule network. The current gradient along with the residual layers are arguments passed to the backward function.

```
1   def backward(x_grad, layers, gamma=0.9):
2       x, v = load() # load from forward step
3       v_grad = initialize()
4       for layer in reversed(layers):
5           x = x - v
6           f_eval = layer(x)
7           v += -(1 - gamma) * f_eval
8           v /= gamma
9           grad = x_grad+v_grad
10          x_grad, v_grad = compute_grad(f_eval, x, v, grad)
11      return x_grad, v_grad
```

## 4 Experimental evaluation

We evaluated MoCapsNet on four different datasets and compared it to other capsule network models. In Section 4.1, we give a detailed description of our setup and the used datasets, and we finalize summarizing our results.

Table 1: Test accuracy of capsule networks with increasing depth (1-8 residual blocks) on MNIST and SVHN. One block consists of two capsule layers. Shown are the average values and standard deviation over three runs. We highlight the best accuracies for each dataset in bold font.

| | MNIST | | SVHN | |
|---|---|---|---|---|
| Blocks | ResCapsNet | MoCapsNet | ResCapsNet | MoCapsNet |
| 1 | 99.41 ± 0.01 | 99.42 ± 0.04 | 92.32 ± 0.52 | 92.68 ± 0.23 |
| 2 | 99.28 ± 0.05 | 99.25 ± 0.11 | 92.57 ± 0.12 | 92.54 ± 0.43 |
| 3 | 99.30 ± 0.08 | 99.31 ± 0.04 | 92.13 ± 0.59 | **93.00 ± 0.65** |
| 4 | 99.38 ± 0.03 | 99.42 ± 0.05 | 91.87 ± 0.95 | 92.03 ± 1.12 |
| 5 | 99.30 ± 0.03 | 99.27 ± 0.06 | 92.58 ± 0.42 | 92.78 ± 0.81 |
| 6 | 99.35 ± 0.02 | 99.38 ± 0.02 | 92.37 ± 0.16 | 92.50 ± 1.60 |
| 7 | 99.36 ± 0.02 | **99.54 ± 0.23** | 92.52 ± 0.23 | 91.35 ± 1.60 |
| 8 | 99.34 ± 0.06 | 99.37 ± 0.06 | 92.63 ± 0.34 | 91.20 ± 1.58 |

### 4.1 Setup & Hyperparameters

Our implementation is implemented in Pytorch (Paszke et al., 2019) and is publicly available on GitHub [2]. The weight of the Momentum term was set to $\gamma=0.9$. We initialize the weights of the transformation matrices at random from a normal distribution with mean 0 and standard deviation 0.01. The batch size for training was 128 and we trained each model for 30 (MNIST) or 60 (SVHN, CIFAR-10/100) epochs. We optimized our network weights with ADAM (Kingma & Ba, 2014), using an initial learning rate of $10^{-3}$ and an exponential decay of 0.96. We use 32 capsules in each capsule layer that is located inside a residual block. The shallowest ResCapsNet and MoCapsNet consist of one residual block, in either case each block is composed of 2 hidden capsule layers (Figure 2). Residual blocks are added to the classical Capsule Network, such that the shallowest ResCapsNet and MoCapsNet will consist of four capsule layers: First capsule layer, 1 block (= 2 hidden capsule layers) and output capsule layer (Figure 1).

In the *PrimaryCapsule* layer, we do a convolution with a kernel size of 9 and a stride of 2. Afterwards, we do a reshaping into capsule form. All following capsule layers (*CapsLayer 1*, capsule layers inside residual blocks, *CapsLayer 2*) share the same setup. Each layer is called with the same routing algorithm, the same number of routing iterations, and has the same length. All layers have 32 capsules, except layer CapsLayer 2, which has the same number of capsules as there are classes in the dataset.

We evaluated our model on four different, popular datasets: MNIST (LeCun et al., 2010), SVHN (Netzer et al., 2012), CIFAR-10 and CIFAR-100 (Krizhevsky, 2009). CIFAR-100 contains images of 100 different classes, the other datasets contain ten classes each. We preprocess images by padding two pixels along all borders and take random crops of size $28 \times 28$ (MNIST) or $32 \times 32$ (SVHN, CIFAR-10/100). After cropping, we normalized per image to have zero mean and a variance of 1. We did not apply any data augmentation techniques.

### 4.2 Results

Table 1 shows the test accuracies of ResCapsNet and MoCapsNet at various depths for MNIST and SVHN, averaged over three runs. Likewise, table 2 shows test accuracies for CIFAR-10 and CIFAR-100. CapsNet and ResCapsNet use the same model architecture, with the only difference being that ResCapsNet uses shortcut connections between capsule layers (see Figure 2a), while CapsNet does not use residual learning. We do not include CapsNet in such table since CapsNet did not achieve better results than chance in any case at any depth (Gugglberger et al., 2021). On the other hand, we obtained a consistently good performance for deeper configurations (more blocks) when using residual shortcut connections, be either ResCapsNet (Gugglberger et al., 2021) or MoCapsNet. The exception was for CIFAR-100, where only MoCapsNet provided good results, and what is even more interesting, the deeper the capsule network, the better the results. Each block of ResCapsNet and MoCapsNet consists of 2 hidden layers added between the first capsule layer and

---

[2]https://redacted

Table 2: Test accuracy of capsule networks with increasing depth (1-8 residual blocks) on CIFAR-10 and CIFAR-100. One block consists of two capsule layers. Shown are the average values and standard deviation over three runs. We highlight the best accuracies for each dataset in bold font.

| | CIFAR-10 | | CIFAR-100 | |
|---|---|---|---|---|
| Blocks | ResCapsNet | MoCapsNet | ResCapsNet | MoCapsNet |
| 1 | $71.49 \pm 0.39$ | $\mathbf{72.18 \pm 0.62}$ | $\approx 1.00$ | $9.53 \pm 0.04$ |
| 2 | $70.59 \pm 0.90$ | $71.65 \pm 0.62$ | $\approx 1.00$ | $26.05 \pm 0.12$ |
| 3 | $71.09 \pm 0.86$ | $71.08 \pm 0.71$ | $\approx 1.00$ | $33.24 \pm 0.29$ |
| 4 | $71.50 \pm 0.57$ | $70.29 \pm 0.06$ | $\approx 1.00$ | $39.57 \pm 0.11$ |
| 5 | $71.94 \pm 0.34$ | $71.74 \pm 0.74$ | $\approx 1.00$ | $42.54 \pm 0.16$ |
| 6 | $71.22 \pm 0.70$ | $71.17 \pm 0.32$ | $\approx 1.00$ | $42.79 \pm 0.13$ |
| 7 | $71.85 \pm 0.91$ | $71.50 \pm 0.93$ | $\approx 1.00$ | $43.22 \pm 0.17$ |
| 8 | $70.90 \pm 1.02$ | $70.48 \pm 0.69$ | $\approx 1.00$ | $\mathbf{43.48 \pm 0.06}$ |

the output layer (Figure 1). Deeper configurations led to better results in MNIST (7 blocks = 14 hidden layers) and SVHN (3 blocks = 6 hidden layers). The accuracy of MoCapsNet and ResCapsNet are almost the same for MNIST. For the case of SVHN and CIFAR-10, MoCapsNet achieves the best accuracy (72.18 %) for CIFAR-10 with 1 residual block, as well as for SVHN (93.00 %) with 3 residual blocks. On CIFAR-100 our deepest model (8 residual blocks) performed best, with an accuracy of 43.48 %. On the other hand, with this dataset – much more complex than its simplified version CIFAR-10 – ResCapsNet was not able to learn the 100-class recognition task at all. We can see here the real benefit of using deeper capsule networks with momentum, where the other capsule network models seem to have collapsed. With CIFAR-100 we also see the large increase in performance when using deeper models, compare the case of 8 blocks (43.48 %) as opposed to only one (9.53 %) or two (26.05 %) blocks.

For the sake of completeness, we compare ResCapsNets and MoCapsNets with the CapsNet implementations from Xi et al. (2017) in Table 3. This implementation reaches an accuracy of 68.93% on CIFAR-10, but thanks to two incremental improvements – compare CapsNet, CapsNet (2-Conv) and CapsNet (2-Conv + 4-ensemble) –, an accuracy of 71.50% can be reached. These improvements consist first of the use of two convolutional layers in front of the capsule networks, and second through training a 4-model ensemble. Even so, our model MoCapsNet (using the same optimizer and routing) with one residual block beats (72.18%) the best CapsNet (71.50%) without the need for an ensemble model. ResCapsNet's accuracy was similar to the ensemble CapsNet model. However, Sabour et al. (2017) reached an error rate of 10.6 % with an 7-model ensemble, but due to resource limitations we were not able to reproduce those results.

We chose CIFAR-10 for this analysis since it was the only dataset (Table 2) where we did not obtain higher accuracies when increasing depth by more than one block (i.e. two added hidden capsule layers between the first capsule layer and output layer). For this case, we further analyzed if there can be accuracy gains when increasing the depth of deeper (more blocks) capsule networks under different routing and optimization. Table 3 compares updates of CapsNet (Xi et al., 2017) with ResCapsNet and MoCapsNet on CIFAR-10. There are two ways we can improve MoCapsNet for better behavior on deeper architectures: Changing the optimizer and the routing algorithm. If we switch our optimizer from ADAM (Kingma & Ba, 2014) to the state-of-the-art optimizer Ranger21 (Wright & Demeure, 2021), the latter handles better depth, reaching 75.15% accuracy on CIFAR-10 with three blocks, improving the results over ResCapsNet (74.24%) and MoCapsNet with one block (74.78%). Secondly, we switch our routing algorithm from routing-by-agreement (RBA) (Sabour et al., 2017) to scaled-distance-agreement (SDA) (Peer et al., 2018), a routing algorithm designed for a better encoding of part/whole relationships. SDA routing again increased the performance, leading to our best performing model on CIFAR-10, consisting of 5 blocks and providing a 75.46% accuracy, compared with the accuracy of 73.67% for one block using the same routing and optimizer. Table 3 shows the benefit of using deeper capsule networks also in CIFAR-10 when using a more novel optimizer or a more recent routing algorithm. We can also see that the change of the forward rule done in the MoCapsNet improves the models' performance when we compare it against the ResCapsNet architecture in all cases.

Table 3: Comparison of our best performing models with a baseline capsule networks on CIFAR-10. NA = does Not Apply, as CapsNets do not include Residual blocks.
* Values from Xi et al. (2017)

| Model | Blocks | Optimizer | Routing | Accuracy |
|---|---|---|---|---|
| CapsNet | NA | Adam | RBA | 68.93 % * |
| CapsNet (2-Conv) | NA | Adam | RBA | 69.34 % * |
| CapsNet (2-Conv + 4-ensemble) | NA | Adam | RBA | 71.50 % * |
| ResCapsNet | 1 | Adam | RBA | 71.49 % |
| MoCapsNet | 1 | Adam | RBA | 72.18 % |
| ResCapsNet | 1 | Ranger21 | RBA | 73.90 % |
| MoCapsNet | 1 | Ranger21 | RBA | **74.78 %** |
| ResCapsNet | 1 | Adam | SDA | 73.16 % |
| MoCapsNet | 1 | Adam | SDA | 73.67 % |
| ResCapsNet | 1 | Ranger21 | SDA | 73.94 % |
| MoCapsNet | 1 | Ranger21 | SDA | 72.87 % |
| ResCapsNet | 3 | Adam | RBA | 71.09 % |
| MoCapsNet | 3 | Adam | RBA | 71.08 % |
| ResCapsNet | 3 | Ranger21 | RBA | 74.24 % |
| MoCapsNet | 3 | Ranger21 | RBA | **75.15 %** |
| ResCapsNet | 3 | Adam | SDA | 74.02 % |
| MoCapsNet | 3 | Adam | SDA | 74.91 % |
| ResCapsNet | 3 | Ranger21 | SDA | 73.90 % |
| MoCapsNet | 3 | Ranger21 | SDA | 73.53 % |
| ResCapsNet | 5 | Adam | RBA | 71.94 % |
| MoCapsNet | 5 | Adam | RBA | 71.74 % |
| ResCapsNet | 5 | Ranger21 | RBA | 74.49 % |
| MoCapsNet | 5 | Ranger21 | RBA | 73.47 % |
| ResCapsNet | 5 | Adam | SDA | 74.16 % |
| MoCapsNet | 5 | Adam | SDA | **75.46 %** |
| ResCapsNet | 5 | Ranger21 | SDA | 69.07 % |
| MoCapsNet | 5 | Ranger21 | SDA | 72.63 % |

Table 4: Memory consumption in MB of capsule networks with increasing depth (1-5 residual blocks) when training on MNIST, CIFAR-10 and SVHN. One block contains two capsule layers. (Res)CapsNet are the results for both, CapsNet and ResCapsNet.

| Dataset | Method | 1 | 2 | 3 | 4 | 5 | 6 | 7 | 8 |
|---|---|---|---|---|---|---|---|---|---|
| MNIST | (Res)CapsNet | 4462 | 4612 | 4858 | 5199 | 5204 | 5502 | 5572 | 5786 |
| | MoCapsNet | 4320 | 4324 | 4328 | 4332 | 4336 | 4546 | 4552 | 4554 |
| CIFAR-10 | (Res)CapsNet | 7605 | 7865 | 8127 | 8197 | 8139 | 8399 | 8437 | 9179 |
| | MoCapsNet | 7653 | 7657 | 7660 | 7663 | 7667 | 7671 | 7675 | 7679 |
| SVHN | (Res)CapsNet | 7326 | 7586 | 7763 | 7918 | 7860 | 8120 | 8158 | 8900 |
| | MoCapsNet | 7371 | 7375 | 7379 | 7381 | 7385 | 7389 | 7393 | 7397 |

The main goal of our MoCapsNet is the ability to train deeper capsule networks, which is hindered by the large memory requirements of capsule networks (section 1). We show the memory consumption in Table 4 (and figs. 3a, 3d and 3g). Values shown are the average over three runs. The memory consumption for MoCapsNet increases by only 4 MB when adding a new residual block. On the other hand, adding a residual block to ResCapsNet increases the memory footprint by around 185 MB, which is 45 times the memory increase of MoCapsNet. While our MoCapsNet needs roughly the same amount of memory as ResCapsNet when using one residual block, if we make the model deeper, for example, with 8 residual blocks, MoCapsNet requires 500 MB less for CIFAR-10 and 1500 MB less for SVHN as its non-reversible counterpart ResCapsNet.

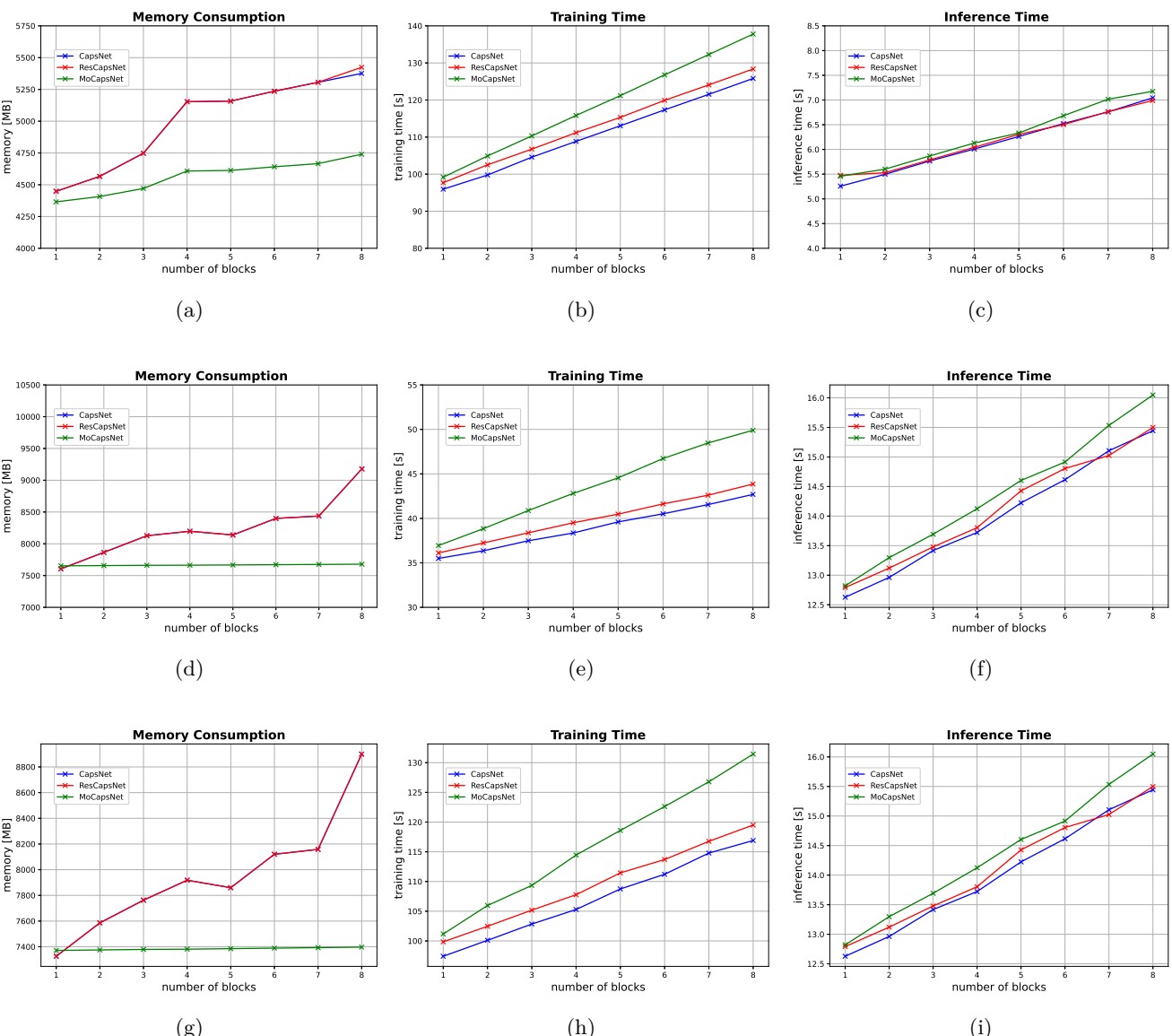

Figure 3: Comparison of memory consumption (first column: (a), (d), (g)), training time (second column: (b), (e), (h)), and inference time (third column: (c), (f), (i)) between CapsNet (blue), ResCapsNet (red), and MoCapsNet (green) with increasing network depth on MNIST (first row, (a) to (c)), CIFAR-10 (second row, (d) to (f)), and SVHN (third row, (g) to (i)). As training and inference time does not vary much across epochs, we report values of the last epoch only.

Note that the memory consumption of CapsNet and ResCapsNet are the same because residual connections do not contain learnable parameters. For the sake of completeness, we also trained a very deep MoCapsNet with 20 blocks (=40 layers) on SVHN and got a stable accuracy of 91.67%.

The price we pay for saving memory can be seen on figures 3b, 3e, 3h. The required time for training the network has a steeper increase (roughly by a factor of 1.8) when using the partly invertible architecture of our MoCapsNet due to re-computations of neuron activations in the backward pass. Inference time is on par, even though it was slightly higher when using momentum (just $0.5s$ in the worst case, i.e., approx. 5% higher), as figures 3b, 3e, and 3h show, which is the result of the new forward rule. This new forward rule requires some small additional computation due to multiplying the momentum term $\gamma$ with the velocity of the previous layer and the activations of the current layer (see lines 4-5 in Algorithm 1). For the sake of

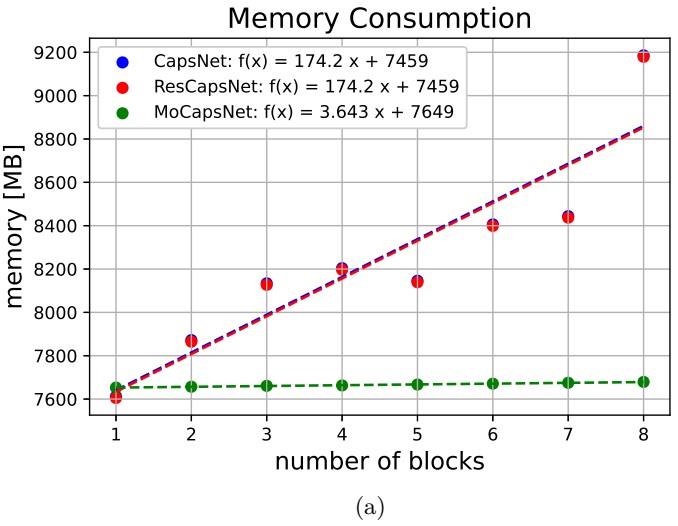
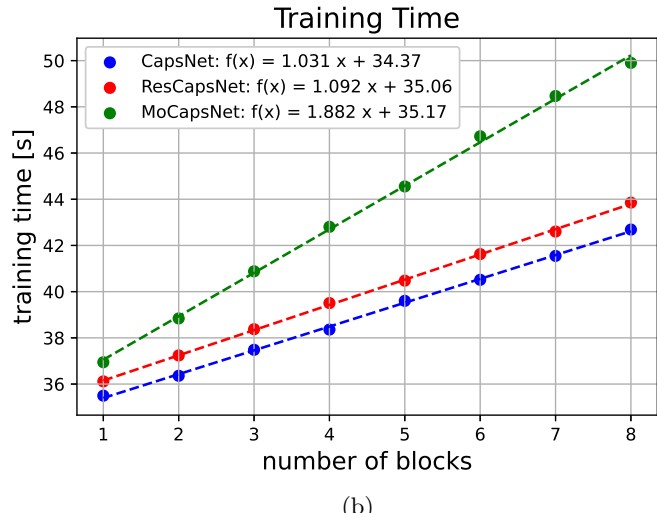

(a)

(b)

Figure 4: Comparison of memory and training time on CapsNet, ResCapsNet and MoCapsNet for CIFAR-10 with an increasing number of residual blocks (one block consists of two layers). The legend shows the memory consumption (a) or training time (b) as a linear function of the number of blocks.

completeness, we analyze deeper the effect on training time compared to memory savings in Figure 4. The memory requirements as the capsule network increases in depth are much flatter in the case of MoCapsNet than CapsNet or ResCapsNet, the steepness of memory as a function of the number of blocks (1 block = 2 layers) for the last two is 174.2, while for MoCapsNet is just 3.643, that is, memory increase over depth is almost 48 times less for MoCapsNet than for CapsNet or ResCapsNet. Even though the training time was steeper in the case of MoCapsNet, the difference with CapsNet and ResCapsNet was small when considering the memory differences. Training time as a function of depth is shown in Figure 4b, adding momentum meant just 1.8 times the computational load (steepness of 1.882 for MoCapsNet vs 1.031 for CapsNet).

We finalize our evaluation by analyzing different hyperparameter values which deviate from the usual setup (section 4.1). Our basic model has 32 capsule types in each capsule layer that is located inside just one residual block. Our evaluation includes different batch sizes, learning rates and the number of capsules in the residual blocks. In order to evaluate the effect of momentum under those hyperparameter variations, our baseline model here for comparison is a residual capsule network (ResCapsNet) with two residual blocks, a batch size of 128, an initial learning rate of 0.01, optimized with Ranger21 (Wright & Demeure, 2021), and trained over 100 epochs.

We compare ResCapsNet and MoCapsNet in Table 5. Neither a higher or lower batch size nor a higher or lower learning rate has a big impact on the accuracy of the model. MoCapsNet achieved better results than ResCapsNet in all cases except for the case of a higher learning rate of 0.1, which had a higher negative impact in MoCapsNet than in ResCapsNet. It is also worth noting that increasing the number of capsules in the layers of the residual blocks to 64 has caused some performance gain in ResCapsNet, but also it has a very high memory consumption of about 14.6 GB, whereas ResCapsNet with 32 capsules per layer needs only 7.9 GB.

## 5 Conclusions & Future Work

We have introduced in this paper Momentum Capsule Networks (MoCapsNet), a new capsule network architecture that implements residual blocks using capsule layers, which can be used to construct reversible building blocks. Through the use of reversible subnetworks, we have obtained a network that has a much smaller memory footprint than its non-invertible counterpart. This fact would allow for the training of capsule networks at much deeper configurations than current setups, thanks to a much more reduced memory

Table 5: Comparison of ResCapsNet and MoCapsNet on CIFAR-10 for different learning rates (0.01, 0.001 and 0.1), number of capsules (32 and 64) in the residual blocks and batch sizes (64, 128 and 256).

| Model | learning rate | number capsules | batch size | accuracy |
|---|---|---|---|---|
| ResCapsNet | 0.01 | 32 | 128 | 74.85 % |
| MoCapsNet | 0.01 | 32 | 128 | **75.26 %** |
| ResCapsNet | 0.01 | 32 | **64** | 72.02 % |
| MoCapsNet | 0.01 | 32 | **64** | **73.85 %** |
| ResCapsNet | 0.01 | 32 | **256** | 74.03 % |
| MoCapsNet | 0.01 | 32 | **256** | **74.13 %** |
| ResCapsNet | 0.01 | **64** | 128 | 74.83 % |
| MoCapsNet | 0.01 | **64** | 128 | **75.52 %** |
| ResCapsNet | **0.001** | 32 | 128 | 72.70 % |
| MoCapsNet | **0.001** | 32 | 128 | **73.14 %** |
| ResCapsNet | **0.1** | 32 | 128 | **70.84 %** |
| MoCapsNet | **0.1** | 32 | 128 | 68.98 % |

consumption. The trade-off of reversible architectures is an increased training time. In spite of that, our experimental evaluation shows that the reduction of memory consumption shrinks at much larger rates than the increase in training times, which added to similar inference time, shows the advantage of MoCapsNet over capsule networks not using reversible building blocks. Our results show that the memory requirements as the capsule network increases in depth are much flatter (Figure 4a) in the case of MoCapsNet than for CapsNet or ResCapsNet, the steepness of memory as a function of the number of blocks (1 block = 2 layers) for MoCapsNet is 48 times less steep than for CapsNet or ResCapsNet. On the other hand, the added computation time when compared with CapsNet and ResCapsNet was quite small (Figure 4b) if we consider such great memory savings. This clear benefit of MoCapsNet over non-reversible architectures is what allows for the training of deeper capsule networks where the current limiting factor is memory. In terms of performance, we have shown experimentally that the modification of the forward rule leads to much improved results of the capsule network. MoCapsNet provides even better results in terms of accuracy on MNIST, SVHN, and CIFAR-10 than recent improved CapsNets consisting of ensembles of networks (Table 3).

Similarly to what happens for CNNs, deeper capsule networks configurations performed better in most cases than flat ones. Recent optimizers such as Ranger21 (Wright & Demeure, 2021) led to even better results on deeper MoCapsNets. Similarly, routing algorithms designed for deeper architectures such as scaled-distance-agreement (SDA) provided higher accuracy values than using the classical routing-by agreement (RBA). In fact, our best performing model on the quite capsule-network-challenging dataset CIFAR-10 consisted of 5 residual blocks deep and using SDA routing, with a 75.46 % accuracy. For completeness, we included a hyperparameter analysis that evaluated different values of learning rates, number of capsules and batch sizes, increasing the number of capsules had a positive effect on MoCapsNet, while changing the batch size or learning rate did not have a big influence or worsened the results.

For future work, it would be of interest to analyze replacing the fully connected capsule layers with convolutional capsule layers, which would enable us to save even more memory. Such modification could allow for the application of capsule networks to more complex tasks and/or datasets, such as ImageNet.

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
