# OpenReview forum: "Momentum Capsule Networks"
_TMLR — Accepted by TMLR_

### Review · Reviewer_fsDs · 2022-04-19

**Summary Of Contributions:**

This work aims at tackling the high-memory requirement of capsule network. It proposes MoCapsNet, a neural network parameterization that combines  capsule and momentum residual network. Momentum residual network is a reversible architecture that allows to recompute the network activations during the backward pass and discards the need of storing them.

 The momentum capsule network is evaluated on CIFAR10, SVHN and MNIST datasets. Empirical evaluation shows that MoCapsNet reduces the memory requirements at the expense of computation. MoCapsNet also outperforms previous capsule network baselines such as CapsNet and ResCaspNet.



**Broader Impact Concerns:**

No specific concerns.

**Requested Changes:**

I would encourage the authors to either add experiments that supports better the claims made in the introduction or to precise those claims, specifically regarding that MoCapsNet allows the training of almost any arbitrarily deep configuration.

**Strengths And Weaknesses:**

*Strengths:*
- The paper clearly demonstrates that MoCapsNet allows to reduce the memory requirement of capsule networks at the expense of  computation.
- MoCapsNet performs better than other capsule network baselines (ResCapsNet and CapsNet).
- Authors say that they will release the code to ease reproducibility of the results.

*Weaknesses:*
- Imprecise claim: Authors claim that they can "train capsule networks at almost any arbitrarily deep configuration" and argue that "the bottleneck for training deeper networks is memory".  However they only considers MoCaspNet of 8 blocks max in the experiments. Depth usually introduces optimization difficulty. It is unclear if MoCapsNet eases the optimization of very deep capsule network.

- Clarity could be improved:  What is the parametrization of PrimaryCaspules and CapsLayer 1/2 in Figure 1? In particular,  it was unclear that the paper was focusing on fully connected capsule from my first read. Similarly,  what is margin loss in equation (1) ? Could you extend on the sentence that "the shortcut connection is inserted after the routing process is completed,

- The memory/compute tradeoff of reversible architectures has been explored as highlighted in the related work section.  The specific application of reversible architecture to capsule network seems novel.

Questions:
- Why do you report the training time only of the last epoch in Figure 3?

---

> ### Author Response · Authors · 2022-05-31
> **Reply**
>
> First, thanks for your valuable feedback. In the following comment, we explain what we did in order to address the concerns of your review. We will upload the new version of our paper soon.
>
> > Imprecise claim: Authors claim that they can "train capsule networks at almost any arbitrarily deep configuration" and argue that "the bottleneck for training deeper networks is memory". However, they only considers MoCaspNet of 8 blocks max in the experiments. Depth usually introduces optimization difficulty. It is unclear if MoCapsNet eases the optimization of very deep capsule network.
>
> Even though, 8 blocks (=16 layers) is already a very deep architecture in the context of capsule networks, we agree with the reviewer that we should provide stronger support for that claim. For this reason, we added another experiment to the revised version of the paper (in section '4.1 Results') where we trained a MoCapsNet with 20 blocks (=40 layers) on SVHN and got a stable accuracy of 91.67%, showing that - to the best of our knowledge - we can train much deeper capsule networks than the current state-of-the-art.
>
> > Clarity could be improved: What is the parametrization of PrimaryCaspules and CapsLayer 1/2 in Figure 1? In particular, it was unclear that the paper was focusing on fully connected capsule from my first read. Similarly, what is margin loss in equation (1) ? Could you extend on the sentence that "the shortcut connection is inserted after the routing process is completed.
>
> We have addressed those points to make the paper more clear, as follows:
> - We now discuss the parametrization of capsule layers in section '4.1 Setup & Hyperparameters'.
> - We now mention fully connected capsule layers in section '3 Momentum Capsule Networks' and in section '3.1 Momentum Residual blocks'.
> - The margin loss was presented by (Sabour et al., 2017). For clarity, we added a sentence explaining it in section '3 Momentum Capsule Networks'.
> - We re-formulated the aforementioned sentence, and consider it is more clear now.
>
> > Why do you report the training time only of the last epoch in Figure 3?
>
> The training and inference time do not vary much across epochs. To clarify how we created measurements, we included this information in Figure 3.

---

> > ### Comment · Reviewer_fsDs · 2022-07-12
> > **Thank you for your response**
> >
> > Thanks for your response. The claim appears correct to me and I agree that the presentation has been improved.

---

### Review · Reviewer_QSyL · 2022-05-03

**Summary Of Contributions:**

This paper proposes Momentum Capsule network, a new network architecture design that combines the capsule network with momentum residual modules. In particular they insert momentum residual modules in between the capsule layers. This results in memory saving benefits since the momentum residual modules are reversible. The resulting MoCapNet (using momentum residual connections) is also marginally better than ResCapNet (using residual connections).

**Requested Changes:**

- The main concern of the paper is that the memory saving story isn't particularly strong. If the author could show that with this technique they can train a very deep CapsNet with much stronger performance, then it would be more impactful. However, Table 5 suggests that the performance already plateaued with 5 blocks.

**Strengths And Weaknesses:**

Strengths:
- The paper did a thorough investigation comparing CapsNet, ResCapsNet and MoCapsNet on several datasets using different hyperparameter settings.

Weakness:
- The combination of momentum residual connection and capsule network might not be particularly interesting to investigate. On one hand, the reversibility of momentum residual networks can give memory saving benefits for many network architectures. On the other hand, capsule networks do not seem to be particularly memory intensive. In Figure 3, all of the memory consumption curves are well under 10GB, which can be handled by modern GPU hardware.
- The paper did a good job explaining the momentum residual connections, but for completeness it should also describe what is inside f_caps.
- The differences between ResCapsNet and MoCapsNet is small (I don’t think it should be the focus of the paper anyways since the performance is expected to be comparable from the momentum residual net paper).

---

> ### Author Response · Authors · 2022-05-31
> **Reply**
>
> First, thanks for your valuable feedback. In the following comment, we explain what we did in order to address the concerns of your review. We will upload the new version of our paper soon.
>
> > The combination of momentum residual connection and capsule network might not be particularly interesting to investigate. On one hand, the reversibility of momentum residual networks can give memory saving benefits for many network architectures. On the other hand, capsule networks do not seem to be particularly memory intensive. In Figure 3, all of the memory consumption curves are well under 10GB, which can be handled by modern GPU hardware.
>
> We clarified in the revised version of the paper (section '1 Introduction'), that memory consumption is currently one major bottleneck for capsule networks: More precisely, each capsule of layer l computes a vote for each capsule of layer l+1. Therefore, the memory consumption grows quadratically with the number of capsules for each layer. On the other hand, our work on Momentum capsule networks keeps the memory consumption constant w.r.t. its depth (Figure 4), which allows for the training of very deep networks. Related to this latter point, we show in the revised version of the paper that we can train networks with up to 40 layers. Please note that capsule networks are usually trained with 2 layers and deeper capsule networks is extremely rare. One exception could be Rajasegaran et al. (2016) with 16 layers, but even in this case, from those layers, *only one* capsule layer exists in their architecture (see https://github.com/brjathu/deepcaps/issues/15), unlike our deep capsule networks where in addition to being able to train a much deeper network (up to 40 layers), except for the first convolutional layer, the rest are *all* capsule layers.
>
> > The paper did a good job explaining the momentum residual connections, but for completeness it should also describe what is inside f_caps.
>
> In the revised version of the paper, we describe the setup of the capsule layers in more detail in section 4.1.
>
> > The differences between ResCapsNet and MoCapsNet is small (I don’t think it should be the focus of the paper anyways since the performance is expected to be comparable from the momentum residual net paper).
>
> I terms of performance, we agree that there is a small improvement of MoCapsNet over ResCapsNet, but the main advantage of Momentum capsule networks is on memory consumption w.r.t depth (Figure 4). We consider this aspect of relevance for TMLR as it may enable future research for the training of much deeper capsule networks.
>
> > The main concern of the paper is that the memory saving story isn't particularly strong. If the author could show that with this technique they can train a very deep CapsNet with much stronger performance, then it would be more impactful. However, Table 5 suggests that the performance already plateaued with 5 blocks.
>
> While our work doesn't provide new state-of-the-art performance, we show a way to overcome the memory bottleneck that comes with deep capsule networks, which we find very relevant for TMLR as the focus is on the notion of "interest" and  "not on impactful w.r.t performance or achieving a new state-of-the-art on some benchmark". We believe, that MoCapsNet could be of particular interest for researches working with - and trying to apply new problems to - capsule networks, which through future routing strategies, finetuning, etc. could reach much higher performance levels through deeper capsule networks as the memory requirements for such networks is much reduced thanks to our work.

---

> > ### Comment · Reviewer_QSyL · 2022-07-12
> > **Thank you for your response**
> >
> > Thanks for your response and it addressed some of my concerns. I agree that the paper is technically correct and could potentially be interesting to the community, even though the performance hasn't been shown to be strong enough to be considered in practice.

---

### Review · Reviewer_xYkD · 2022-05-18

**Summary Of Contributions:**

This paper proposes a new architecture, Momentum Capsule Networks, to address the high memory consumption required for training Capsule Networks. The key contribution is to borrow techniques in reversible networks to recompute activations in the backward pass, so that activations do not need to be saved.

This submission evaluates the proposed network on three standard image classification benchmarks, MNIST, SVHN, and CIFAR10. The proposed network improves the performance of Capsule Networks and reduce the training memory consumption. However, the proposed network was not shown to reach state of the art results (in either the line of capsule network research or considering other convolutional network architecture).

Overall, I find the submission to be a direct application of existing techniques (reversible network) to Capsule network, which means that the submission is less novel. I find the presentation to be confusing and not well-organized at times and provide suggestions in the following sections. In summary, I recommend rejection of the submission.

**Broader Impact Concerns:**

I do not have ethical concern with the submission.

**Requested Changes:**

## Requested Changes

- The first paragraph of the introduction should be about the main claim of this paper. Currently, it described many low-level details about the capsule network, which makes this paragraph very long and hard to read
- The second paragraph of the introduction mentioned three problems of the capsule networks but this paper really only touches upon the last one. I suggest to cut the first two and focus on what this paper is trying to solve.
- The second paragraph of the introduction conflates two challenges in training deeper capsule networks: instability and memory consumption. This paper only aims at solving the latter. I suggest the authors to remove the discussion on instability.
- The third paragraph of the introduction provides an overview of the results. I recommend the authors to cite concrete numbers to showcase the benefits of the proposed methods.
- Section 2.1, “capsule networks require vast amounts of memory.” is not supported by the following example “… consisting of 14.36 million parameters”. The claim is about memory consumption during training and the example is about the parameter count. These are two different concepts
- Section 2.3, unclear how this line of work is relevant to the submission
- Section 3, first paragraph, the claim about “requires significant more memory during training” needs to be supported by actual numbers.
- I suggest to reorganize Section 3 into subsections: 3.1 background on capsule network; 3.2 background on reversible network; 3.3. proposed method.
- Section 3.1, “The design of the block is very similar to a common residual building block used in ResNet architectures, but such blocks need to be adapted in order to handle fully connected capsule layers.” The paper should explain the reason for why there is such a “need to be adapted”.
- Section 3.1, “Differently from the classical ResNet architecture,” → “Different from”.
- I suggest to formalize the descriptions in text and figure 1 into equations.;
- Algorithm 1 and 2 are now named “Listing” in captions.
- Section 4.2, “However, Sabour et al. (2017) reached an error rate of 10.6 % with an 7-model ensemble, but due to resource limitations we were not able to reproduce those results.” This is very unsatisfying. Can the author explain what kind of resource would be required to reproduce such results?
- Figure 3 is very confusing. The authors should 1. show legends 2. mark different plots with the network depth. Also the plots on different rows look very similar and I am not sure there is much value in duplicating them many times.
- “The required time for training the network has a steeper increase when using the partly invertible architecture of our MoCapsNet due to re-computations of neuron activations in the backward pass.” please provide actual numbers describing “stepper increase”.

**Strengths And Weaknesses:**

## Strength

- The proposed method was able to reduce memory consumption of training capsule networks, which is the main motivation for this submission
- The proposed method shows performance improvement over the capsule network

## Weakness

- Proposed method is not novel.
- Paper could be organized in better form and writing needs to be improved. I provide suggestions in the next section.

---

> ### Author Response · Authors · 2022-06-01
> **Reply**
>
> First, thanks for your valuable feedback. In the following comment, we explain what we did in order to address the concerns of your review.
>
> > Proposed method is not novel.
>
> We respectfully disagree with this unjustified statement - something which TMLR would disapprove according to its reviewer guidelines -, as this is the first time Momentum is implemented in Capsule Networks. Even though, we are unsure for the reasons behind such strong negative statement, we have tried to make our contributions more clear along the paper. The research effort leading to the derivation of Momentum Capsule Networks is explained in section 3. We show the interest of our work as it has strong implications for training deep capsule networks (section 4), one of the limiting factors of using such architectures.
>
> > The first paragraph of the introduction should be about the main claim of this paper. Currently, it described many low-level details about the capsule network, which makes this paragraph very long and hard to read
>
> We agree that the introduction is about the main claim of the paper, although for clarity we think it needs some preliminary context. We have revised such paragraph to make it more concise and get to the contributions earlier in this section.
>
> > The second paragraph of the introduction mentioned three problems of the capsule networks but this paper really only touches upon the last one. I suggest to cut the first two and focus on what this paper is trying to solve.
> The second paragraph of the introduction conflates two challenges in training deeper capsule networks: instability and memory consumption. This paper only aims at solving the latter. I suggest the authors to remove the discussion on instability.
>
> We adapted this paragraph following the reviewer suggestions while trying to also satisfy reviewer fsDs and added experimental evaluation on"deeper networks to show more that "MoCapsNet eases the optimization of deep capsule networks". We hope that this additional experimentation addresses the concerns of both reviewers.
>
> > The third paragraph of the introduction provides an overview of the results. I recommend the authors to cite concrete numbers to showcase the benefits of the proposed methods.
>
> In the revised paper, we now include the most important results in this section, as suggested.
>
> > Section 2.1, “capsule networks require vast amounts of memory.” is not supported by the following example “… consisting of 14.36 million parameters”. The claim is about memory consumption during training and the example is about the parameter count. These are two different concepts
>
> We agree with the reviewer and have removed the parameter count from this section as it may be misleading.
>
> > Section 2.3, unclear how this line of work is relevant to the submission
>
> The residual learning framework is fundamental for our work (see section 3.1.). Without residual learning, the momentum architecture would not be possible.
>
> > Section 3, first paragraph, the claim about “requires significant more memory during training” needs to be supported by actual numbers.
>
> We have clarified in the revised version of the paper (section '1 Introduction'), that memory consumption is currently one major bottleneck for capsule networks: More precisely, each capsule of layer l computes a vote for each capsule of layer l+1. Therefore, the memory consumption grows quadratically with the number of capsules for each layer, which would support this claim in section 3.
>
> > I suggest to reorganize Section 3 into subsections: 3.1 background on capsule network; 3.2 background on reversible network; 3.3. proposed method.
>
> We agree with such structure, although we address those subsections in the "related work" section in order to keep section 3 only for the methodology.
>
> > Section 3.1, “The design of the block is very similar to a common residual building block used in ResNet architectures, but such blocks need to be adapted in order to handle fully connected capsule layers.” The paper should explain the reason for why there is such a “need to be adapted”.
>
> We have clarified this aspect and added "The layers need to be adapted because they represent capsules (not neurons), which perform dynamic routing with upper level capsules".
>
> > Section 3.1, “Differently from the classical ResNet architecture,” → “Different from”.
>
> Fixed in the new version.
>
> > I suggest to formalize the descriptions in text and figure 1 into equations.;
>
> We have included the equations corresponding to the contributions of our work (Momentum residual blocks). The rest of the layers are common capsule network layers.
>
> > Algorithm 1 and 2 are now named “Listing” in captions.
>
> Fixed in new version.

---

> > ### Comment · Reviewer_xYkD · 2022-07-13
> > **Response to reply**
> >
> > I have looked at the revised draft and believe that the writing has been improved.
> >
> > I mostly agree with other reviewers. I think the main claims made in this paper are technically correct. I maintained my original position on novelty but also agree that novelty is not the focus on TMLR.

---

> ### Author Response · Authors · 2022-06-01
> **Reply 2**
>
> > Section 4.2, “However, Sabour et al. (2017) reached an error rate of 10.6 % with an 7-model ensemble, but due to resource limitations we were not able to reproduce those results.” This is very unsatisfying. Can the author explain what kind of resource would be required to reproduce such results?
>
> Training an ensemble of 7 models is computationally very expensive with our setup, as we execute 3 runs for each experiment in order to report the mean and std-dev. This would result in 21 training runs for a single experiment. As the value is already reported in Sabour et al. (2017) we do not think it adds any value to replicate such expensive experimental evaluation. We have removed "but due to resource limitations we were not able to reproduce those results" for clarity.
>
> > Figure 3 is very confusing. The authors should 1. show legends 2. mark different plots with the network depth. Also the plots on different rows look very similar and I am not sure there is much value in duplicating them many times.
>
> Each sub-figure in Figure 3 has their own legend and network depth labels in the revised manuscript.
>
> > “The required time for training the network has a steeper increase when using the partly invertible architecture of our MoCapsNet due to re-computations of neuron activations in the backward pass.” please provide actual numbers describing “stepper increase”.
>
> For clarity, we mention the increasing factor in the text in the revised version of the paper. Actual values corresponding to the increase can be found in Figure 4.

---

### Comment · Action_Editors · 2022-05-20
**Discussion period has begun**

Dear authors and reviewers,

Now that all three reviews are in, the discussion period has officially begun, where authors are encouraged to address the reviewers concerns.
Authors: do not hesitate to reach out for any questions. I believe the reviews are of high-quality and point to very specific aspects of improvement.

Thanks!
AE

---

### Decision · Action_Editors · 2022-07-07

**Recommendation:** Accept with minor revision

**Comment:**

This work addresses the memory bottleneck that comes with deep capsule networks, by investigating an architectural variant based on reversible residual blocks, or `Momentum ResNets'. The authors carry out experiments on several mid-scale benchmarks, demonstrating improvements in memory consumption and performance relative to other capsule-based models.

The three experts reviewers who assessed this work agreed that this work does provide memory and performance improvements over other capsule networks, enabling training of deeper architectures. They also noted that, in its current form, this paper does not fully capitalize on this observation, and that the submission would be much more convincing if the authors could leverage the memory savings to train a large-scale capsule network with strong empirical performance on a larger-scale dataset. The AE agrees with this assessment. In conclusion, according to the TMLR guidelines we are happy to accept this manuscript, while strongly encouraging the authors to make a final effort to include these experiments.

---

> ### Author Response · Authors · 2022-07-19
> **Additional Experiments**
>
> We are very thankful for the acceptance of our paper. We are following the recommendation to do such experiments and will include results on larger datasets on the final version of the paper.